# Unraveling the Complexity of Vaso-Occlusive Crises in Sickle Cell Disease: Insights from a Resource-Limited Setting

**DOI:** 10.3390/jcm13092528

**Published:** 2024-04-25

**Authors:** Ali Kaponda, Kalunga Muya, Jules Panda, Kodondi Kule Koto, Bruno Bonnechère

**Affiliations:** 1Reference Centre for Sickle Cell Disease of Lubumbashi, Institut de Recherche en Science de la Santé, Lubumbashi 1825, Democratic Republic of the Congo; alik@unilu.ac.cd (A.K.); jupand@hotmail.com (J.P.); 2Department of Clinical Biology, Faculty of Pharmaceutical Sciences, University of Lubumbashi, Lubumbashi 1825, Democratic Republic of the Congo; 3Department of Pharmacology, Faculty of Pharmaceutical Sciences, University of Lubumbashi, Lubumbashi 1825, Democratic Republic of the Congo; kalunga.muya@unilu.ac.cd; 4Department of Surgery, Faculty of Medicine, University of Lubumbashi, Lubumbashi 1825, Democratic Republic of the Congo; 5Department of Clinical Biology, Faculty of Pharmaceutical Sciences, University of Kinshasa, Kinshasa 2212, Democratic Republic of the Congo; k_kodondi@yahoo.fr; 6REVAL, Rehabilitation Research Center, Faculty of Rehabilitation Sciences, University of Hasselt, 3590 Hasselt, Belgium; 7Technology-Supported and Data-Driven Rehabilitation, Data Science Institute, University of Hasselt, 3590 Hasselt, Belgium; 8Department of PXL—Healthcare, PXL University of Applied Sciences and Arts, 3500 Hasselt, Belgium

**Keywords:** sickle cell disease, vaso-occlusive crises, resource-limited setting, hematological parameters, patient care

## Abstract

**Background/Objectives**: This study investigated vaso-occlusive crises (VOCs) in sickle cell disease in Lubumbashi, Democratic Republic of Congo, aiming to understand the disease complexities amidst limited resources. With sickle cell hemoglobinopathies on the rise in sub-Saharan Africa, this nine-year study explored factors associated with VOCs and hematological components. **Methods**: This study comprised 838 patients, analyzing VOCs and hematological changes over time. Demographic characteristics and blood composition changes were carefully categorized. A total of 2910 crises were observed and managed, with analyses conducted on severity, localization, and age groups using statistical methods. **Results**: The majority of crises were mild or moderate, primarily affecting osteoarticular regions. Statistical analysis revealed significant disparities in crisis intensity based on location and age. The association between blood samples and the number of comorbidities was investigated. Significant positive associations were found for all parameters, except monocytes, indicating a potential link between blood variables and complication burden. Survival analysis using Cox regression was performed to predict the probability of experiencing a second crisis. No significant effects of medication or localization were observed. However, intensity (*p* < 0.001), age (*p* < 0.001), and gender (*p* < 0.001) showed significant effects. Adjusted Hazard Ratios indicated increased risk with age and male gender and reduced risk with mild or severe crisis intensity compared to light. **Conclusions**: This research sheds light on the complexities of VOCs in resource-limited settings where sickle cell disease is prevalent. The intricate interplay between clinical, laboratory, and treatment factors is highlighted, offering insights for improved patient care. It aims to raise awareness of patient challenges and provide valuable information for targeted interventions to alleviate their burden.

## 1. Introduction

The escalating incidence of sickle cell hemoglobinopathies is most pronounced in sub-Saharan Africa, where over 300,000 newborns with sickle cell disease are born annually, constituting approximately 1% of the region’s total births [1,2,3].

Sickle cell disease is a severe hereditary form of anemia. It results from a single point mutation of an adenine to a thymine, leading to the synthesis of an abnormal hemoglobin (HbS). In its deoxygenated state, HbS is able to polymerize, causing damage in numerous organ systems and leading to various clinical manifestations of varying severity from one patient to another, despite the genetic identity of the site of the sickle cell hemoglobin mutation [4].

The pathogenesis of vaso-occlusive crisis (VOC) in sickle cell disease is complex, encompassing various cellular and molecular interactions. Inflammation in SCD, a hallmark of sickle cell pathology, which triggers a series of events that lead to cell adhesion and aggregation., is driven by multiple factors, including the release of pro-inflammatory cytokines and the activation of adhesion molecules on endothelial cells. These inflammatory mediators create a highly adhesive environment, promoting the interaction between sickle cells, endothelial cells, platelets, and other blood components. Sickle cells, with their altered shape and reduced flexibility, are more prone to adhering to blood vessel walls, further contributing to vaso-occlusion. This cascade leads to vaso-occlusion and intense acute pain [5], affecting multiple organs [3,6,7]. 

The pain associated with VOC is described variably as throbbing, cutting, gnawing, or generalized toothache and is the predominant reason for medical consultation and hospital admissions [1,8,9,10]. Beyond the acute pain, VOCs can lead to long-term complications as multiple organs become affected due to repeated ischemic episodes. The cascade of events during VOC is central to understanding the broader implications of sickle cell disease, highlighting the need for targeted therapies that can interrupt or mitigate these complex interactions. 

Therefore, individuals with sickle cell disease exhibit diverse VOCs and/or related complications, often not fully comprehended by clinicians. The frequency of these painful crises is significantly correlated with blood biological fluctuations [11,12,13,14,15]. Given the primarily palliative nature of sickle cell disease and its resultant VOCs, the management necessitates the utilization of a multitude of pharmacological agents [16,17].

This study was conducted within a resource-limited setting with the aim of enhancing patient care by elucidating the clinical, laboratory, and therapeutic predictors of painful VOCs in their environment. Our focus was on observing parameters related to VOCs (ages, gender, intensity, and localization) and several standard hematological elements (hemoglobin, sedimentation rate, and blood count) during various crisis types [14,15]. To achieve this, we meticulously describe the types of VOCs encountered in sickle cell subjects over approximately 9 years. We compiled a comprehensive list of observed complications and proposed medications, while also attempting to delineate predictive factors for VOC occurrence based on clinical and biological profiles, along with the proposed medication, considering various factors present in resource-challenged environments.

## 2. Materials and Methods

### 2.1. Environment

This retrospective study was conducted at the ‘*Centre de Référence de la Drépanocytose de Lubumbashi*’ (CRDL), located in the Haut-Katanga province of the Democratic Republic of Congo. The CRDL is situated within the premises of the Jason Sendwe Provincial General Referral Hospital, also known as the Jason Sendwe Hospital. This study encompassed all individuals with sickle cell disease who sought consultation at the center from June 2014 to December 2022.

As a provincial branch of the ‘*Institut de Recherche en Sciences de la Santé*’ (IRSS), the CRDL was chosen as the research setting due to the substantial number of patients it serves and its primary mission of addressing the healthcare needs of individuals with sickle cell disease in a resource-limited environment. The prevalence of patients and the center’s dedicated focus on sickle cell disease were influential criteria guiding its selection for investigating vaso-occlusive crises experienced by patients in a disadvantaged setting.

### 2.2. Population

This study encompassed patients of all age groups who sought care at the healthcare facilities managed by the CRDL. Specifically, the focus was on individuals with sickle cell disease who manifested one or more VOCs during the study period. Excluded from this study population were individuals declared non-sicklers and those with sickle cell disease who did not experience any VOCs throughout the study period.

### 2.3. Data Collection

To collect socio-demographic data, hematological parameters, and details of medical management, we utilized patient consultation forms.

The information contained in the consultation forms facilitated the establishment of anthropometric characteristics, clinical manifestations, implemented medical interventions, and hematological aspects of the study population.

To describe the anthropometric characteristics of the study participants, age and gender were considered. Age was categorized into four major groups according to the Sergeant classification: (i) newborn class [0–0.5] years, (ii) early childhood [0.5–5] years, (iii) late childhood [5–15] years, and (iv) adults [15–50] years [18].

Regarding the clinical manifestations presented by the study population, we considered the localization (abdominal, osteoarticular, or mixed (patients presenting both abdominal and osteoarticular complaints during the same crisis)) and intensity (mild, moderate, or severe) of VOCs. The combination of each location with the painful sensation provided 9 modalities.

VOCs are generally associated with other comorbidities. These comorbidities presented by the study patients were grouped into several compartments based on the affected organs and/or systems, including osteoarticular, digestive, Ear, Nose, and Throat (ENT), parasitic and infectious, cardiovascular, respiratory, genitourinary, nephrology, musculo-cutaneous lesions, and gynecological disorders.

To assess the hematological status of the study patients, we examined the erythrocyte sedimentation rate, hemoglobin levels, and white blood cell counts. For the latter, the leukocyte formula was presented in terms of the percentages of lymphocytes, monocytes, and neutrophils.

To describe the received medical management, we presented the therapeutic classes of medications administered by clinicians during various consultations conducted throughout the study period. The different therapeutic classes were formed based on the activities of the molecules, the system involved, the primary indication, etc.

### 2.4. Statistical Analysis

Descriptive statistics, including median age and interquartile range, were used to summarize patient demographics. The time between the first visit and the first crisis was calculated for each patient. Chi-squared tests were conducted to examine associations between crisis severity, localization, and patient age.

Complications associated with crises were analyzed by quantifying the average number of complications per crisis. Chi-squared tests were used to assess differences in complication distribution across crisis severity levels. Linear regression was employed to investigate the influence of age and crisis severity on the number of complications.

Prescription patterns for different drugs were analyzed, with a focus on anti-inflammatory drugs. Regression analysis was performed to determine associations between medication use and patient age, crisis severity, localization, and comorbidities.

Associations between blood variables and the total number of complications were assessed using appropriate statistical methods, such as correlation analysis or linear regression.

Finally, we performed survival analysis, specifically Cox regression, to predict the probability of experiencing a second crisis. The effects of medication, crisis intensity, age, and gender on survival were examined, and Hazard Ratios with corresponding confidence intervals were calculated.

Statistical analyses were performed in R, and the level of significance was set to 0.05.

## 3. Results

### 3.1. Patients and VOCs

A total of 818 patients were included in this study (413 female and 405 male), with a median age of 10 years old (p25 = 5–p75 = 17). According to the Sergeant classification, we had 5 patients under 5 months, 211 patients between 5 months and 5 years, 350 patients between 5 and 15 years, and 252 patients aged above 15 years old. The median time between the first visit and the next crisis was 1.07 years (p25 = 0.12–p75 = 3.21).

During the follow-up period, 2910 crises were monitored. Amongst these, the majority were classified as mild (46.7%, n = 1360) and light (45.2%, n = 1316), and severe crises were found in 8.0% of the cases (n = 234). Concerning the localization, the vast majority of the crises were osteoarticular (59.5%, n = 1730), followed by mixed (27.2%, n = 793), and abdominal forms were found in only 13.2% of the crises (n = 387). Table 1 presents the distribution of the crises according to severity and localization; the chi^2^ test shows important differences between the localization and the intensity (*p* < 0.001), the most notable difference being that most of the osteoarticular crisis is of moderate intensity, while abdominal ones lead to light symptomatology. We also analyzed the age of the participants in the different groups and found statistically significant differences according to localization (*p* < 0.001) and severity (*p* < 0.001) but no interaction (*p* = 0.43).

### 3.2. Complications

We then analyzed the number of complications associated with the crisis (Table 2); on average, the patients had two complications per crisis. The chi-squared test revealed differences in the distribution of the complications according to the severity of the crisis. Notably, there is a linear trend for the prevalence of blood complications associated with the severity of the crisis (only 10% found in light crises and 37.5% for severe ones). Interestingly, para-infectious complications were less frequent in severe crises (74.9%) in comparison with light and mild crises (86.3 and 86.6%, respectively).

We then performed linear regressions to determine the influence of age and the severity of the crisis on the number of complications. There was a significant and negative effect of age on the number of complications (β = −0.37, SE = 0.08, *p* < 0.001) and severity (β = 2.04, SE = 0.03, *p* < 0.001 for mild compared to light and β = 5.29, SE = 0.61, *p* < 0.001 for severe compared to light), see Figure 1.

Then, we analyzed the relationship between the localization and severity on the number of complications. We found a significant effect of the localization (*p* < 0.001) and severity (*p* = 0.024) and an interaction between these two (*p* = 0.014). In comparison with abdominal crises, the osteoarticular crises, on average, led to fewer complications (β = −1.28, SE = 0.11, *p* < 0.001). There was a significant interaction between the mild intensity and mixed localization (β = −0.41, SE = 0.13, *p* = 0.044), as well as between mild intensity and osteoarticular localization (β = −0.66, SE = 0.21, *p* = 0.002).

### 3.3. Medications

An important aspect of the management of the crises is the medication; therefore, we analyzed the (poly)medication according to severity, localization, and age. First, we present the results of the use of different drugs according to the severity of the crises. Overall, by far, the most frequently used drugs, regardless of the severity of the crises, are the anti-inflammatory drugs (1.5 prescriptions per crisis, on average); the complete results are presented in Figure 2 and Table 3.

We then performed regression to determine if there is an effect of the age, severity, localization, and comorbidities. There was no significant effect of age (*p* = 0.21). We found a significant association with a linear trend for the severity (β = 1.15, SE = 0.10, *p* < 0.001 for mild compared to light and β = 1.89, SE = 0.19, *p* < 0.001 for severe compared to light) (Figure 3). Concerning the localization, the mixed forms were more medicated than the abdominal ones (β = 1.20, SE = 0.16, *p* < 0.001), and the osteoarticular were less medicated than the abdominal ones (β = −0.38, SE = 0.15, *p* = 0.014). Finally, concerning the comorbidities, we found a positive effect of the total number of comorbidities (β = 0.51, SE = 0.03, *p* < 0.001), as well as an interaction with the total number of comorbidities and the severity of the crisis (β = 0.14, SE = 0.05, *p* = 0.005 and β = 0.43, SE = 0.10, *p* < 0.001 for the interaction between comorbidities and mild intensity and severe intensity, respectively).

### 3.4. Biology

An important question was to determine if there are associations between blood samples and the number of comorbidities. As presented in Table 4, for all parameters, except the monocytes, we found a significant and positive association between the blood variables and the total number of complications.

### 3.5. Survival Analysis

Finally, we performed a survival analysis to predict the probability of having a second crisis. The results of the Cox regression show that there is no effect of the medication (*p* = 0.86) or the localization (*p* = 0.19). On the other hand, significant effects were found for the intensity (*p* < 0.001), age (*p* < 0.001), and sex (*p* < 0.001) (Figure 4). We did not find a significant interaction between these parameters. Therefore, the adjusted Hazard Ratio (HR) values with their corresponding 95% confidence intervals were 0.97 [0.97–0.98] for age, 1.15 [1.07–1.24] for gender, with males being at increased risk, 0.80 [0.74–0.86] for intensity for mild severity compared to light, and 0.81 [0.70–0.93] for severe compared to light.

## 4. Discussion

### 4.1. Demographic Data

Similarly, as observed in other studies, this investigation revealed that VOCs manifest in various ways among individuals with sickle cell disease [19]. A notable predominance of females was observed among patients who developed VOC. However, this gender predominance cannot be justified due to the autosomal non-sex-linked transmission of the disease. With a median age of 10 years, the majority of the studied patients were young. Approximately 75% of them were under 17 years old, with one-third of them being under 5 years old. The prevalence of young patients can be attributed to several factors, including (i) parental decision-making regarding their attendance at consultations, (ii) the short life expectancy of individuals with sickle cell disease, resulting in fewer elderly patients, (iii) older patients, who are few in number and often in despair, possessing a better understanding of their condition, managing it more effectively, and seeking medical attention only when the situation deteriorates, and (iv) the predominant youth population in Africa [5,20,21,22].

### 4.2. VOC’s Type

Sickle cell disease is associated with significant morbidity stemming from both acute and chronic complications, leading to organ dysfunction as early as the first year of life [23]. VOC represents one of the major acute complications of sickle cell disease, with its prevalence varying across age groups and influencing various comorbidities from early childhood to adulthood [10,24,25]. Similar to the findings by Platt and colleagues, our results indicate that age exerts a negative influence on the mild severity of VOC and the number of associated complications, particularly in the context of mild versus moderate severity and severe intensity versus mild intensity [26]. Note that in resource-limited settings, the older the child, the more he or she develops experience and a certain memory of the crisis and tends to manage the crisis alone. These older patients rarely come for check-ups or treatment, nor do they respect the golden rules of their sickle cell lifestyle. This favors falciformation and, consequently, the onset of increasingly severe VOCs. In children and infants, on the other hand, basic management is administered and monitored by the parents, which is thought to reduce the occurrence of VOCs. Older children, adolescents, and adults are also exposed to various types of stress. This, too, is likely to favor the onset of VOC.

Recurrent episodes of VOC have a significant impact on cumulative organ damage [7], with age emerging as a key determinant not only in terms of pain localization and intensity but also in the occurrence of other complications associated with VOC.

Pain localization and intensity play a fundamental role in sickle cell disease. Osteoarticular VOC predominated (59.5%), likely attributed to the ease of perception of bone pain. Mild-intensity VOC can be managed easily by patients at home [17] as individuals tolerate mild pain more readily, often considering it transient and occasional, with the tendency to become persistent with age, sometimes complicating diagnosis [27,28]. Moderate-intensity VOC accounted for a significant portion (46.7%), consistent with observations elsewhere that moderate-intensity VOCs are more commonly encountered [29]. Moderate osteoarticular crises were notably more prevalent than mild abdominal crises, possibly due to the ease of confusing abdominal pain with various other digestive disorders [30].

### 4.3. Infections

In sub-Saharan countries, without any healthcare intervention, 50 to 90% of children with sickle cell disease will succumb in childhood, primarily due to infections that may progress to septicemia [31,32]. Interestingly, infectious complications were more prevalent during moderate crises (86.5%) compared to mild ones (86.3%) and were less frequently observed in patients experiencing severe crises (74.9%). This observation aligns with the slower progression of infections compared to other complications such as hemolysis, which can be more rapidly debilitating [33]. The lower frequency of para-infectious complications in severe CVO could be explained by the fact that patients with severe pain are less cooperative during the anamnesis and clinical examination. As a result, severe CVOs are quickly managed to alleviate pain, and at the same time, anti-infectious and anti-parasitic treatment is often associated with them. This is performed routinely, especially in resource-limited environments where hygiene is precarious and patients are exposed to multiple infections and parasitosis.

Bacterial infections and severe complications of sickle cell disease remain frequent. The prevalence of complications, in descending order, includes parasitic and infectious diseases (86.6%), digestive issues (43.7%), pulmonary complications (25.7%), anemias (18.2%), and ENT (ear, nose, and throat) disorders (17.5%). Infectious manifestations are common in all these groups of pathologies, except for digestive and anemic complications.

### 4.4. Medications

Given the highly variable clinical expression of sickle cell disease, various therapeutic strategies have been employed. Prescriptions encompassed 187 molecules administered individually or in combination, grouped into 31 pharmacological classes used to manage VOC and associated comorbidities. This classification was based on the symptoms addressed and the therapeutic goals. The management of numerous complications associated with VOC relies on symptomatic medication tailored to age, severity, and localization [34,35].

Nonsteroidal anti-inflammatory drugs (NSAIDs) and opioids are effective in alleviating acute pain, with guidelines available to aid healthcare providers in managing sickle cell pain [36]. Throughout this study, regardless of the severity of crises, NSAIDs were consistently the most prescribed medications (an average of 1.5 prescriptions per crisis). This underscores the significant inflammatory imbalance in VOC, with less utilization of anticoagulants, antiseptics, and diuretics, primarily employed for less common complications. In contrast, the predominant use of opioids has been observed elsewhere. For mild or moderate VOC, treatment typically begins with NSAIDs, while opioids, the first-line treatment for more intense pain, should be administered immediately [37]. Opioids are favored over NSAIDs because they rapidly alleviate pain, although they may pose challenges such as respiratory failure, particularly in resource-poor settings lacking adequate equipment [25].

In the therapeutic arsenal encountered, opioids are rarely administered parenterally, mainly due to a lack of expertise and the absence of intensive care facilities. Medical training is limited to day hospitals, potentially increasing the use of NSAIDs, especially as patients resort to self-medication and only seek hospital care in case of treatment failure. This partly explains why NSAIDs were the most widely used medications, regardless of crisis severity (1.5 prescriptions per crisis on average). Moreover, their usage was proportional to the severity of the crisis. However, NSAID complications, such as varying degrees of gastric mucosal damage leading to gastritis or gastric ulcers, may contribute to the high prevalence of digestive complications (43.7%), given the considerable stress associated with sickle cell disease [38,39].

The management of mixed VOC involves a greater number of medications than that of osteoarticular or abdominal VOC. This complexity arises because mixed VOC often necessitates the combination of medications to alleviate pain in various organs or systems affected [40].

An increase in the number of comorbidities has a positive effect on crisis severity. Each complication or comorbidity contributes to the overall severity of the crisis [41,42,43]. Several studies have demonstrated that penicillin prophylaxis during the first five years and the curative use of antibiotics contribute to the high prevalence of antibiotic use (81.1%) and antibiotic–antiparasitic combination (13.4%) [44,45].

Hydroxyurea, an orally administered medication with a well-established safety and efficacy profile, induces a reduction in morbidity and mortality in patients with sickle cell disease [46]. This treatment is recommended from the age of 9 months and is indicated for both pediatric and adult patients with sickle cell disease [47,48]. Hydroxyurea improves chronic organ damage and prolongs survival by reducing residual hemoglobin, which protects against the intracellular polymerization of HbS [49].

Although Olupot’s work notes a 50% reduction in malaria incidence among sickle cell children in sub-Saharan Africa with hydroxyurea, these beneficial effects may not manifest in our environment due to its delayed onset of action and the need for careful biological monitoring [47]. Furthermore, the stringent requirements for accessing hydroxyurea present a challenging reality for patients, especially considering their high susceptibility to malaria and low rates of adherence to biological diagnosis [50]. Additionally, its limited accessibility and/or high cost should not be overlooked [51,52].

### 4.5. Biology

Certain blood parameters are crucial in monitoring sickle cell crises [14]. Among the hematological elements typically requested in our environment, six parameters were most commonly found in the analyzed hemograms: hemoglobin (49.97%), white blood cells (33.61%), sedimentation rate (30.79%), lymphocytes (30.55%), neutrophils (30.52%), and monocytes (15.19%). These six parameters enable the categorization and characterization of anemias while providing insights into the patient’s susceptibility to infections. It is noteworthy that these parameters are often selected when patients are unable to afford the costs associated with a complete blood count. Additionally, these six parameters are easy to measure with simple basic equipment, without necessarily requiring automation.

The findings of Ugwu and colleagues align with ours, indicating significantly higher average values for hematological parameters in individuals experiencing severe crises compared to those with mild to moderate crises [14]. Kamble’s study on individuals with sickle cell SS genotype also demonstrates fluctuating mean hemoglobin levels of 7.8 g% and 6.8 g% in subjects with or without vaso-occlusive crises [53]. Our results are nearly similar, considering a control group comprising the entire patient cohort (*n* = 1454), with an average hemoglobin level of 7.8 g% compared to 6.99 g% during vaso-occlusive crises. Like a previous study [53], we did not demonstrate a direct correlation between hemoglobin and vaso-occlusive crises. However, there was a correlation between the number of complications and hemoglobin levels (*p* = 9.14 × 10^6^), even considering the variation in this level (*p* = 0.04).

In sickle cell disease, there is a high risk of morbidity and mortality associated with malaria, particularly due to severe anemia and increased hemolysis. This may be attributed to the challenging management of blood-related complications, which tend to escalate rapidly [33,54].

Complications in sickle cell disease are diverse, with at least two complications identified during each crisis. Blood-related complications were more prevalent in individuals with severe vaso-occlusive crises. Those with blood disorders had a significantly lower average hemoglobin level compared to the overall average hemoglobin level in individuals with sickle cell disease. This strongly suggests that blood disorders contribute to the destruction of red blood cells.

### 4.6. Predictions of VOCs

Survival analysis did not demonstrate the effect of medication (*p* = 0.86) or location (*p* = 0.19) on the occurrence of a second crisis. Similarly, in other studies, factors such as location, the number of medications taken, age, gender, and biology are not cited as predisposing factors for the occurrence of a second vaso-occlusive crisis [26]. However, the literature mentions hematological parameters such as hematocrit and fetal hemoglobin levels as significant risk factors for vaso-occlusive crises [26]. Contrary to these factors, our results did not identify hematocrit and fetal hemoglobin levels as predictors of a second vaso-occlusive crisis. Instead, our findings are almost similar to those of Platt in 1999, highlighting age and gender as predictive factors for a subsequent crisis [26]. Notably, the risk is higher in males. Considering the intensity, subjects experiencing mild vaso-occlusive crises are 0.8 times less likely to develop a second crisis [7,26,55].

Previous studies have shown that in individuals with sickle cell disease, higher hemoglobin levels are associated with a greater frequency of vaso-occlusive crises. In our analysis, the occurrence of a second crisis, although statistically non-significant, was more likely to be encountered in males than females. This could be explained by the fact that males are generally less anemic than females [17,56].

Mild and moderate intensities, as well as age, non-significantly decrease the occurrence of a second crisis. It is acknowledged that with age, hemoglobin levels decrease in individuals with sickle cell disease, exposing them to other complications (especially cardiac) but reducing the frequency of vaso-occlusive crises. As mentioned earlier, naturally anemic individuals tend to experience fewer vaso-occlusive crises [18]. Patients with low pain levels may easily develop a second crisis as they are often less effectively managed during crises with low pain severity [57,58,59].

### 4.7. Limitations

This work provides useful insights into the management and characteristics of VOCs in sickle cell disease in a setting with limited resources. However, it is important to acknowledge several limitations in order to fully comprehend the context and applicability of the findings.

This study’s retrospective design is based on preexisting data, which may not include all relevant clinical and treatment aspects or the complete range of VOC severity and outcomes. This dependence limits the capacity to offer a thorough comprehension of VOCs and might disregard important variables. The findings of this study may not be applicable to other regions with distinct healthcare infrastructures, processes, or patient demographics due to this study’s specific geographical and healthcare context. The emphasis on a solitary facility presents the possibility of selection bias as the sample may not truly represent the wider demography of patients with sickle cell disease.

Moreover, this study mainly focused on parameters associated with VOCs and alterations in blood composition. Future studies would be enhanced by broadening the scope to incorporate genetic determinants, evaluations of the patient’s quality of life, or long-term therapy outcomes in order to attain a fuller comprehension. In addition, although this study employed patient consultation forms for data gathering, it did not provide comprehensive information on the particular techniques of data collection, measuring tools, and potential biases. Providing a clear and comprehensive description of these methodologies, as well as acknowledging any potential limits in the accuracy or completeness of the data, could enhance our comprehension of this disease in future studies.

This study comprised a sample size of 838 patients, which may be deemed quite limited for making extensive generalizations, particularly given the heterogeneity observed within sickle cell disease populations. Enhancing the sample size or incorporating a wider array of participants could enhance the external validity and generalizability to broader demographics.

This study also emphasized the dependence on analgesics and anti-inflammatory medications for the management of VOCs. Subsequent research could further analyze the efficacy of these medications, examine potential adverse reactions, and seek other therapeutic approaches to improve patient results.

Ultimately, this study recognizes the necessity for additional investigation into biochemical markers and thorough documentation of pharmaceutical side effects, highlighting the fact that our existing knowledge of VOCs in sickle cell disease is still insufficient. To improve the comprehension and treatment of sickle cell disease, it is important to address these constraints in future studies, especially in settings with limited resources.

## 5. Conclusions

This study investigated the hematological profile and medication management of individuals with sickle cell disease during VOCs. Generally, it was observed that individuals experiencing VOCs were predominantly young. Their medication management primarily relied on analgesics and anti-inflammatory drugs, exposing patients to gastrointestinal disturbances. In contrast, elsewhere, the use of opioids, which is favored, helps to avoid such issues. Their hematological profile was characterized by low levels of red blood cells and hemoglobin, as well as elevated levels of leukocytes (monocytes).

Future research could expand upon this work by investigating the biochemical parameters of sickle cell patients and compiling a comprehensive record of the side effects associated with the administered medications. This would contribute to a more thorough understanding of the overall health and treatment outcomes for individuals with sickle cell disease during VOCs.

## Figures and Tables

**Figure 1 jcm-13-02528-f001:**
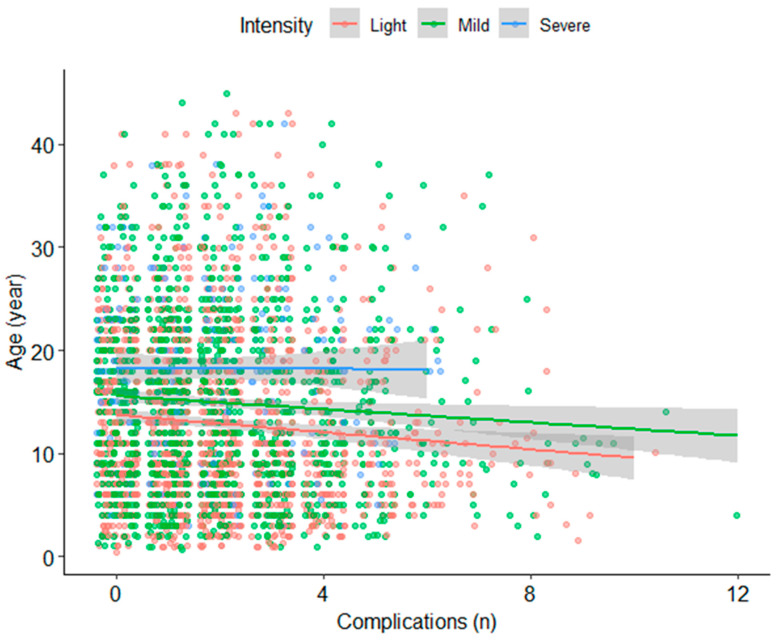
Relationship between the age of the patients and the number of complications.

**Figure 2 jcm-13-02528-f002:**
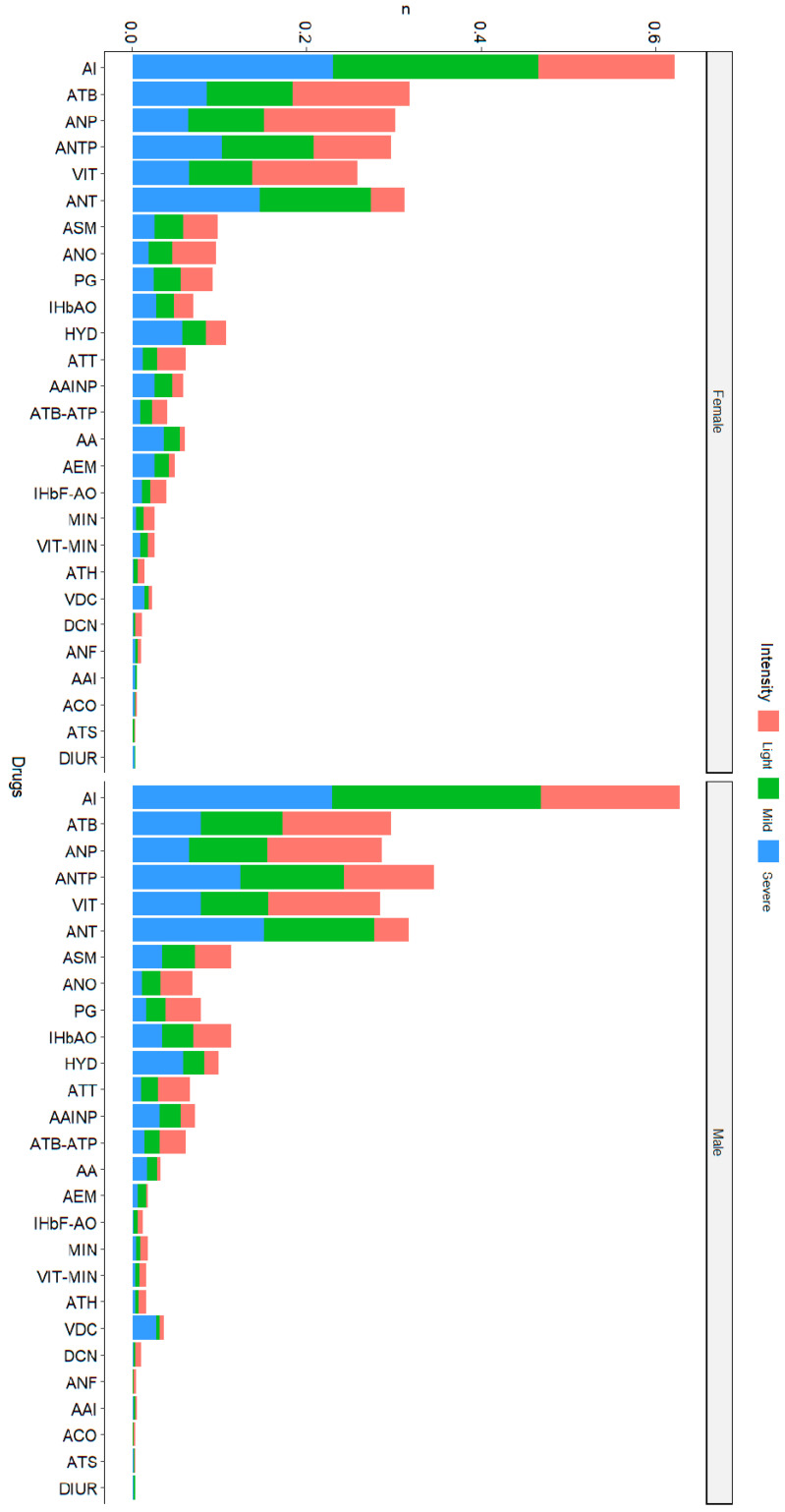
Relationship between VOC severity and type of medication used, according to gender. The list of the medications’ abbreviations is presented in Table 3.

**Figure 3 jcm-13-02528-f003:**
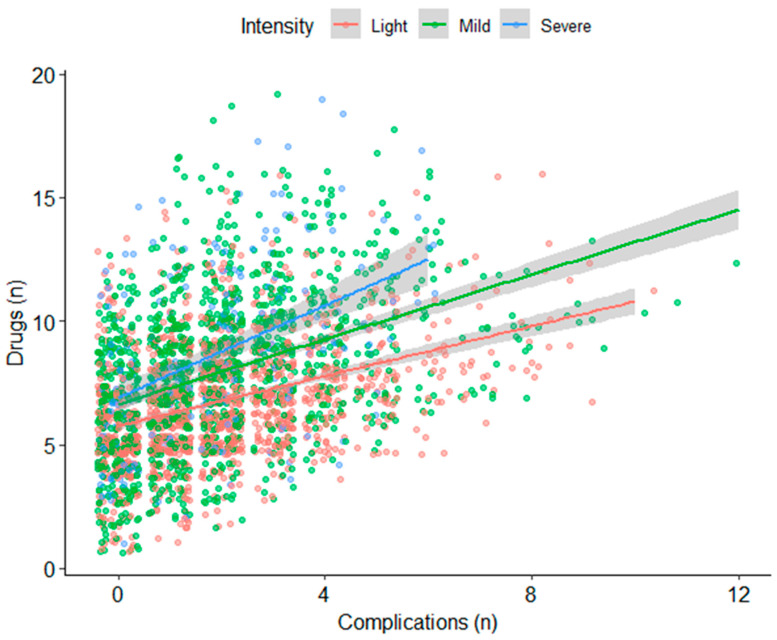
Relationship between the number of complications and the number of medications.

**Figure 4 jcm-13-02528-f004:**
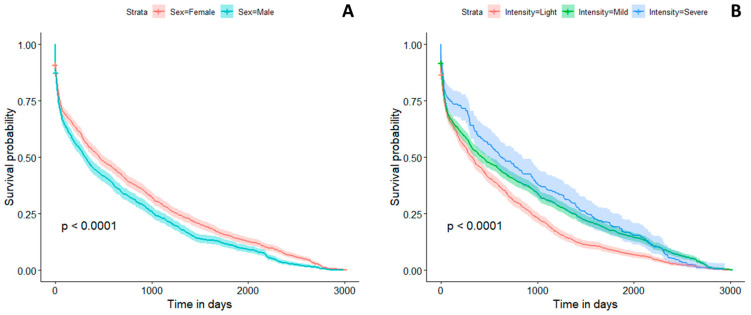
Survival analysis according to gender (**A**) and the severity of VOCs (**B**).

**Table 1 jcm-13-02528-t001:** Distribution of the crises according to severity and localization.

Severity	Localization		
	Osteoarticular	Mixed	Abdominal
Light	*n* = 694 Age = 9.0 [11] Sex ratio = 51.1%	*n* = 337 Age = 9.8 [10] Sex ratio = 52.6%	*n* = 285 Age = 8.7 [8] Sex ratio = 57.6%
Mild	*n* = 872 Age = 11.3 [13] Sex ratio = 55.3%	*n* = 398 Age = 9.6 [13] Sex ratio = 62.9%	*n* = 90 Age = 17.6 [12] Sex ratio = 58.3%
Severe	*n* = 164 Age = 13.7 [8] Sex ratio = 61.2%	*n* = 58 Age = 9.2 [9] Sex ratio = 52.9%	*n* = 12 Age = 6.2 [2] Sex ratio = 33.3%

For the sex ratio, the reference is female. Age is presented as the median [interquartile range].

**Table 2 jcm-13-02528-t002:** Type and prevalence of complications associated with the different severities of crises.

Complications	Overall (*n* = 2910)	Severity		
		Light (*n* = 1316)	Mild (*n* = 1360)	Severe (*n* = 234)
Articular	2.7%	2.1%	3.3%	2.1%
Digestif	43.7%	47.3%	40.2%	37.3%
Ear, Nose, and Throat	17.5%	23.0%	14.3%	4.5%
Para-infectious	86.2%	86.3%	86.6%	74.9%
Musculo-cutaneous	2.5%	3.3%	2.0%	1.2%
Blood	18.2%	10.2%	22.1%	37.5%
Cardiovascular	2.7%	2.1%	3.3%	2.1%
Respiratory	25.7%	28.0%	24.3%	19.3%
Gynecologic	0.6%	0.5%	0.6%	0.4%

Note that neuro, CVD, uro, and nephro were never reported.

**Table 3 jcm-13-02528-t003:** Type of medication associated with the different severities of crises.

Medications (Family)	Overall (*n* = 2910)	Severity		
		Light (*n* = 1316)	Mild (*n* = 1360)	Severe (*n* = 234)
Amino acids (AAs)	9.1%	3.8%	11.5%	24.8%
Analgesic-anti-inflammatory agents (AAIs)	1.4%	0.8%	2.0%	1.7%
Antacids and proton pump inhibitors (AAINPs)	14.4%	9.5%	17.4%	24.8%
Anticoagulants (ACOs)	1.1%	1.1%	1.1%	0.9%
Antiemetic (AEM)	7.4%	3.2%	10.0%	15.4%
Anti-inflammatory (AI)	152.3%	105.6%	188.9%	202.6%
Antifungals (ANFs)	2.1%	2.3%	2.0%	1.3%
Antioxidants (ANOs)	23.5%	29.8%	19.1%	13.7%
Antiparasitics (ANPs)	80.6%	95.6%	70.1%	56.8%
Analgesics (ANTs)	69.7%	26.4%	101.1%	130.3%
Analgesic-antipyretic (ANTP)	78.1%	62.9%	89.3%	98.3%
Antispasmodics (ASMs)	27.4%	26.9%	28.2%	25.2%
Antibiotics (ATBs)	81.1%	87.2%	76.8%	72.6%
Antibiotics and antiparasitics (ATB-ATPs)	13.4%	15.3%	12.1%	9.8%
Antihistamines (ATHs)	4.4%	5.2%	4.0%	1.3%
Antiseptics (ATSs)	1.1%	1.4%	0.9%	0.4%
Antitussives (ATTs)	17.8%	22.9%	14.3%	9.8%
Nasal decongestants (Cos)	3.3%	4.7%	2.4%	0.9%
Diuretics (DIURs)	0.8%	0.3%	1.2%	1.3%
Hydrating agents (HYDs)	19.8%	13.7%	20.4%	51.3%
Hemoglobin inducers and antioxidants (IhbAOs)	21.6%	20.5%	21.8%	26.5%
Fetal hemoglobin inducers and antioxidants (IHBFAOs)	7.1%	8.7%	5.8%	6.0%
Minerals (MINs)	6.1%	7.1%	5.6%	3.8%
Gastric dressing (PG)	23.0%	25.2%	21.7%	17.9%
Vasodilators (VDCs)	4.3%	2.4%	3.9%	17.1%
Vitamins (VITs)	70.3%	83.1%	59.2%	62.8%
Vitamins-minerals (VIT-MINs)	5.7%	5.2%	6.0%	6.0%

**Table 4 jcm-13-02528-t004:** Results of regression between blood variables and the number of complications.

Parameters	*n*	Beta (Complications)	*p*-Value
Hemoglobin	1454	0.18 (0.04)	9.14 × 10^6^
Sedimentation rate	896	0.78 (0.15)	4.55 × 10^7^
Leucocytes	978	517 (88)	5.84 × 10^9^
Lymphocytes	889	0.74 (0.22)	7.24 × 10^4^
Monocytes	442	0.03 (0.02)	0.30
Neutrophils	888	1.13 (0.26)	2.17 × 10^5^

## Data Availability

All of the individual participants’ data collected during this study, after de-identification, are available to researchers who provide a methodologically sound proposal for individual participant data meta-analysis immediately following the publication; no end data. Proposals should be directed to bruno.bonnechere@uhasselt.be to gain access; data requestors will need to sign a data access agreement.

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
