# Peer review of "Unraveling the Complexity of Vaso-Occlusive Crises in Sickle Cell Disease: Insights from a Resource-Limited Setting"

_jcm, 2024, doi:10.3390/jcm13092528_

Round 1
Reviewer 1 Report
Comments and Suggestions for Authors
Based on the study provided, here are some comments that could be addressed or considered in limitations
Limited Scope: The study focused primarily on analyzing factors related to vaso-occlusive crises and hematological changes. To provide a more comprehensive understanding, future research could consider expanding the scope to include other relevant aspects such as genetic factors, patient quality of life assessments, or long-term treatment outcomes.
Data Collection Methods: While the study utilized patient consultation forms for data collection, the details about the specific methodologies used for data collection, measurement tools, and potential biases in data collection were not explicitly mentioned. Clearly outlining the data collection methods and addressing any limitations in data accuracy or completeness would strengthen the study's credibility.
Sample Size and Generalizability: The study included 838 patients, which may be considered relatively small for drawing broad generalizations, especially considering the diversity within sickle cell disease populations. Increasing the sample size or including a more diverse range of participants could enhance the study's external validity and applicability to broader populations.
Medication Management: The study highlighted the reliance on analgesics and anti-inflammatory drugs for medication management during vaso-occlusive crises. To offer a more comprehensive view, future studies could delve deeper into the effectiveness of these medications, potential side effects, and explore alternative treatment strategies to improve patient outcomes.
Reviewer 2 Report
Comments and Suggestions for Authors
The manuscript entitled “Unraveling the Complexity of Vaso-Occlusive Crises in Sickle 2 Cell Disease: Insights from a Resource-Limited Setting” by Kaponda et al., is a well-written retrospective study conducted to investigate the medication management during the vaso-occlusive crises (VOCs) in sickle cell patients in Lubumbashi, Democratic Republic of Congo.
In general, the study was well designed and addresses substantial data on VOCs in SCD pathology and medication for the management of VOCs.
Page 2, line 60: The authors could add a note on the cascade of VOCs when introducing the pathology of the disease. The authors must explain what cascades are involved in the VOCs setting.
Page 4, line 149: Authors have mentioned that they have included 838 patients in the study, however it looks like they had 818 patients in total.
Additionally, it would be interesting for readers if authors included any VOCs they observed in 5 months old patients.
Table 1: It appears that as they age, the severity of VOCs in osteoarticular location rises. Explain.
Page 4, line 172: It would be interesting if authors could explain why the para-infectious complications were less frequent in severe VOCs.
Page 10, line 251: Authors should change CVO’s to VOC’s.
Authors could add a note in VOC’s type if they have observed any vascular complications in severe VOCs patients and what are the medications were prescribed to improve their pain crises.
In addition, the authors have rectify spelling errors like analyze, neutrophils and other grammar errors.
Comments on the Quality of English LanguageAuthors should rectify the grammatical errors when using "analyse" and "neutrophiles".
Reviewer 3 Report
Comments and Suggestions for Authors
In the current manuscript, Ali Kaponda investigates vaso-occlusive crises (VOCs) in sickle cell disease in 23 Lubumbashi, Democratic Republic of Congo, aiming to understand the disease complexities amidst 24 limited resources. However, to further improve the readability of the manuscript, I have the following minor comments:
1. In Table1, the definition of “Localisation Mixed” is vague. Please provide a contxt of this category in either the main text or the table legend.
2. Figure 2: please annotate the unit for x axis.
3. Figure 1 and Figure 3: please provide the equation of the linear regressions in the plot.
Please note that the authors seem to use “analysis” as a verb in the current manuscript instead of “analyze”. Please correct all incidents where the error applies.
Line 175-178, please combine the two sentences to specify what’s exactly the significant effect.
Comments on the Quality of English LanguagePlease note that the authors seem to use “analysis” as a verb in the current manuscript instead of “analyze”. Please correct all incidents where the error applies.
Line 175-178, please combine the two sentences to specify what’s exactly the significant effect.
